# Genome-Wide Identification, Evolution, and miRNA-22 Regulation of *Kruppel-Like Factor* (*KLF*) Gene Family in Chicken (*Gallus gallus*)

**DOI:** 10.3390/ani14172594

**Published:** 2024-09-06

**Authors:** Zheng Ma, Huangbin Chu, Fapei Li, Guochao Han, Yingqiu Cai, Jianing Yi, Mingrou Lu, Hai Xiang, Huimin Kang, Fei Ye, Siyu Chen, Hua Li

**Affiliations:** Guangdong Provincial Key Laboratory of Animal Molecular Design and Precise Breeding, School of Animal Science and Technology, Foshan University, Foshan 528225, China; mz8522@163.com (Z.M.); 17856419337@163.com (H.C.); fapei1525@163.com (F.L.); luminous031116@gmail.com (M.L.);

**Keywords:** chicken, *KLF* gene family, bioinformatic, miR-22

## Abstract

**Simple Summary:**

Krüppel-like factors (KLFs) are essential transcription factors found in many eukaryotes. In this study, the focus is on identifying and analyzing the *KLF* gene family members in chickens using bioinformatic tools, and comparing them with representative classes such as fish, amphibians, birds, and mammals. The analysis includes the structural characterization, evolutionary relationship assessment, and functional predictions. Additionally, in this study, the impact of miRNA-22 is explored, associated with lipid metabolism, on the expression of *KLF* genes in the liver, heart, and muscle tissues of Qingyuan partridge chickens. The results indicate that the avian *KLF* gene family is evolutionarily closer to that of mammals, and all chicken KLFs are non-transmembrane proteins. Moreover, the effect of miRNA-22 on KLF expression varies across different tissues. These findings provide a scientific basis for further research into the functions of KLFs in chickens.

**Abstract:**

Krüppel-like factors (KLFs) are a class of fundamental transcription factors that are widely present in various eukaryotes from nematodes to humans, named after their DNA binding domain which is highly homologous to the Krüppel factor in fruit flies. To investigate the composition, organization, and evolutionary trajectory of *KLF* gene family members in chickens, in our study, we leveraged conserved sequences of *KLF* genes from representative classes across fish, amphibians, birds, and mammals as foundational sequences. Bioinformatic tools were employed to perform homology alignment on the chicken genome database, ultimately identifying the KLF family members present in chickens. The gene structure, phylogenetic analysis, conserved base sequences, physicochemical properties, collinearity analysis, and protein structure were then analyzed using bioinformatic tools. Additionally, the impact of miRNA-22, related to poultry lipid metabolism, on the expression of the *KLF* gene family in the liver, heart, and muscle of Qingyuan partridge chickens was explored. The results showed that: (1) compared to fish, the KLF family in birds is more closely related to mammals and amphibians; (2) KLFs within the same subgroups are likely to be derived from a common ancestral gene duplication; (3) KLF3/8/12 in the same subgroup may have some similar or overlapping functions; (4) the motif 4 of KLF5 was most likely lost during evolution; (5) KLF9 may perform a similar function in chickens and pigs; (6) there are collinear relationships between certain *KLF* genes, indicating that there are related biomolecular functions between these *KLF* genes; (7) all members of the KLF family in chickens are non-transmembrane proteins; and (8) interference and overexpression of miRNA-22 in Qingyuan partridge chickens can affect the expression levels of *KLF* genes in liver, heart, and muscle.

## 1. Introduction

The *KLF* gene family is widely present in eukaryotes and is a transcription factor family. Members of this family are generally present in a zinc finger structure at the carboxy-terminal end, which consists of three consecutive and highly conserved Cys2/His2 residues [1]. This structure enables the KLF transcription factors to bind to GC and CACCC sequences on DNA [2]. So far, 14 members of this family have been identified in poultry [3], and 18 KLF family members have been found in mammals and named KLF1-KLF18 [4].

The *KLF* gene family plays a crucial role in lipid differentiation [5]. Specifically, *KLF5*, *6*, and *KLF15* positively regulate adipocyte differentiation, whereas *KLF2*, *KLF3*, and *KLF7* have a negative regulatory role, and the function of *KLF4* in cell differentiation remains a point of contention. According to Birsoy [6], IBMX can induce *KLF4* expression in mouse 3T3-L1 preadipocytes. When combined with Krox20, *KLF4* activates the expression of C/EBPβ, thereby promoting adipocyte differentiation. However, Park et al. [7] have suggested that *KLF4* does not affect 3T3-L1 preadipocyte differentiation. Concerning KLF transcription factors positively regulating adipogenic differentiation, Xie et al. [8] conducted a temporal and spatial expression analysis of *KLF5* and discovered that the goat *KLF5* gene positively regulates the differentiation of intramuscular adipocytes. This, in turn, promotes intramuscular fat deposition in goats. Similarly, *KLF6* was discovered as promoting adipocyte differentiation by suppressing PREF1 expression [9]. Additionally, the research has shown that the expression of *KLF15* is positively correlated with the process of bovine adipocyte differentiation and *KLF15* can facilitate the accumulation of lipid droplets during adipogenic differentiation [10]. KLF transcription factors can also negatively regulate adipogenic differentiation, with KLF2 inhibiting the activation of PPARγ by binding to its promoter, thereby impeding adipocyte differentiation [11]. KLF3 first binds to CtBP protein to form a KLF3-CtBP inhibitory complex that suppresses C/EBPα expression and thus, inhibits adipocyte differentiation [12]. In addition, KLF7’s role is to hinder adipocyte maturation by inhibiting *GLUT2* gene expression [13].

The above describes the effects of some members of the *KLF* gene family on the differentiation of mammalian adipose tissue. However, there are relatively few reports on the effects of KLF family members on the differentiation of poultry adipocytes. In 2015, Wang et al. [14] found that KLF3 plays an inhibitory role in the process of chicken adipocyte differentiation, which is speculated to be achieved by inhibiting the expression of C/EBPα and *FAS* genes. Subsequently, Wang et al. [15] studied the correlation between *KLF15* gene expression and IMF deposition in Tibetan chicken and found a positive correlation, but the regulatory mechanism needs further study. At the same time, Zhao et al. [16] found that *KLF7* promotes CDKN3 transcription by binding to the binding site “TGGGCGGGCT” (−137/−128) on *CDKN3*, thereby inhibiting chicken preadipocyte differentiation and promoting proliferation. In 2020, Wu et al. [17] successfully constructed a *KLF11* overexpression vector and transfected it into LMH cells. They found that overexpressed *KLF11* promotes LMH cell expression, inhibits INS and ACC expression, and does not affect *FAS* and *SREBP-1c* expression.

MicroRNAs (miRNAs) are small non-coding RNA molecules approximately 22 nucleotides in length that regulate target genes and participate in various biological processes in animals. The research into the miRNAs associated with the *KLF* gene family in poultry has revealed significant findings. For instance, miR-106-5p inhibits the proliferation and adipogenic differentiation of chicken preadipocytes by targeting the *KLF15* gene [18]. Wang et al. [19] found that miR-21 suppresses the proliferation of primary chicken adipocytes by targeting the *KLF5* gene. Additionally, miR-7 has been shown to affect the proliferation and differentiation of primary chicken myoblasts by regulating the expression of the *KLF4* gene [20,21]. miR-22-3p is highly expressed in the liver of laying hens during the egg-laying period, and it participates in lipid metabolism by targeting genes such as *ACSL5*, *ELOVL6*, and *PLIN2* [22]. Li et al. [23] demonstrated that miR-22-3p is involved in hepatic lipid synthesis through the targeted regulation of *ACSL5*. Furthermore, miR-22-3p influences lipid synthesis in the liver by targeting the *ELOVL6* gene [24,25]. Lima et al. [26] discovered that the loss of miR-22 reduces fat accumulation induced by a high-fat diet (HFD). Additionally, Huang et al. [27] reported that miR-22 impacts lipid droplet accumulation and osteoblast differentiation by inhibiting *HDAC6*. From the above, it is evident that chicken miR-22 is closely related to liver lipid metabolism and muscle growth. Coincidentally, the *KLF* gene family is also associated with these physiological processes. Thus, it raises an intriguing question, does chicken miR-22 influence these physiological phenotypes by regulating the expression of the *KLF* gene family?

In this study, we utilized bioinformatic approaches to identify members of the *KLF* gene family within the chicken genome database and analyzed their physicochemical properties. We constructed a phylogenetic tree and examined the gene structure, conserved motifs, synteny relationships, and protein structural characteristics. Additionally, we conducted statistical analyses of *KLF* gene expression in the liver, heart, and muscle tissues of Qingyuan partridge chickens following miR-22-3p lentiviral interference and overexpression. This research aims to elucidate the members and biological characteristics of the *KLF* gene family in chickens and clarify the impact of miR-22-3p on their expression, providing valuable scientific insights for future studies on the biological functions of this gene family in chickens.

## 2. Materials and Methods

### 2.1. Identification and Bioinformatics Analysis of Chicken KLF Gene Family Members

This study initially retrieved the protein sequences of *KLF* gene family members in pigs, tropical clawed frogs, and zebrafish from the UniProt database to establish a fundamental protein library for the *KLF* gene family. The amino acid sequences were then aligned, compared, and saved as FASTA format files using MEGA. The HMMER software (v3.1) was used to find the hidden Markov model (HMM) with these files, and the HMM search was conducted against the chicken protein database to identify potential protein sequences for the *KLF* gene family (E-value < 1 × 10^−10^). Subsequently, Ensembl Blastp was employed to compare and annotate the candidate KLF family amino acid sequences (E-value < 1 × 10^−10^), followed by submitting the protein sequences to the NCBI-CDD website for screening to identify proteins with the complete conserved structural domain of the family. Based on the NCBI annotations, redundant protein sequences were eliminated, and eventually, 15 *KLF* family members were identified in the chicken gene database. The physicochemical properties of the selected chicken KLF family members, including protein molecular weight, amino acid length, and isoelectric point, were analyzed using the ExPASy online software (https://web.expasy.org/protparam/, accessed on 28 May 2022).

### 2.2. Chromosome Location Analysis

This study utilized the Ensembl online platform (http://asia.ensembl.org/Gallus_gallus/Info/Index, accessed on 6 September 2022) to obtain detailed information regarding the chromosomes, their corresponding loci, and the specific positions of each member of the *KLF* gene family. The MapChart online software (https://www.wur.nl/en/show/Mapchart.htm, accessed on 7 September 2022) was then used to conduct a chromosome localization analysis.

### 2.3. Construction of Phylogenetic Tree

The protein sequences of *KLF* gene family members in zebrafish (representative species of fish), tropical clawed frog (representative species of amphibians), chicken (representative species of birds), and pig (representative species of mammals) were downloaded from the NCBI database. The Clustal-w method was utilized with MEGA7 software (v7.0.26) to align these sequences for conducting sequence homology analysis. Subsequently, a phylogenetic tree was constructed using the neighbor-joining (NJ) method with a bootstrap value of 1000.

### 2.4. KLF Gene Structure and Motif Analysis

The GTF file for *KLF* gene family members was downloaded using the online Table Browser (https://genome.ucsc.edu/cgi-bin/hgTables, accessed on 28 September 2022), and the TBtools software was utilized to perform gene structure visualization analysis. The MEME tool (http://meme-suite.org/tools/meme, accessed on 28 September 2022) was used to predict the conserved sequence sites in the chicken KLF protein sequence, with the maximum value set to 10.

### 2.5. Gene Collinearity Analysis

The chicken’s genome sequence was downloaded from the National Center for Biotechnology Information (https://www.ncbi.nlm.nih.gov/, accessed on 22 October 2022). The genes in the *KLF* gene family were mapped using the *GXF* gene Position and info. extract function of TB tools (https://github.com/CJ-Chen/TBtools/releases, accessed on 26 October 2022). The collinear relationship between the selected genes was analyzed by One Step MC Scan X-Super Fast in TB tools. After that, the advanced Circos function of TB tools was used to display the results of gene collinearity.

### 2.6. Prediction of Protein Tertiary Structure and Transmembrane Region

Protein sequences of each member of the *KLF* gene family were downloaded from NCBI, and the SWISS-MODEL online software (https://swissmodel.expasy.org/, accessed on 28 October 2022) was used to predict their protein tertiary structures. The TMHMM online software (https://services.healthtech.dtu.dk/service.php?TMHMM-2.0, accessed on 17 November 2022) was used to predict the transmembrane regions of the protein sequences.

### 2.7. Preparation of miR-22 Lentiviral Vectors, Chicken Treatment Protocols, and Analysis of KLF Gene Family Expression

Construction of lentivirus-mediated miR-22-3p overexpression and interference vectors: The mature sequence of miR-122-5p was first downloaded from miRbase (https://www.mirbase.org/, accessed on 19 November 2022), with the overexpression sequence of miR-22-5p being 5′-AAGCTGCCAGTTGAAGAACTGT. The designed interference sequence was 5′-ACAGTTCTTCAACTGGCAGCTT. These sequences were cloned and inserted into the lentiviral vector pcDNA3.1-exons4-CDS-mCherry, which contains a GFP (green fluorescent protein) reporter gene. The accuracy of the recombinant vector was confirmed by DNA sequencing. The vector was then packaged into a virus using the packaging vectors pMDLg/pRRE, pRSV-Rev, and the envelope protein vector pCMV-VSV-G. Lentiviral packaging was provided by GenePharma (GenePharma, Shanghai, China).

Nine 12-day-old Qingyuan partridge hens from the same generation and batch were selected as subjects (purchased from Guangdong Tinoo Food Group Co., Ltd., Qingyuan, China). The chickens were randomly divided into the following three groups: the miR-22-3p-NC Group: injected with 300 μL of LV3-NC lentivirus at a titer of 10^−9^ TU/mL; the miR-22-3p-I Group: injected with 300 μL of LV3-miR-22-3-inhibit lentivirus at a titer of 10^−9^ TU/mL; and the miR-22-3p-M Group: injected with 300 μL of LV3-miR-22-3-mimic lentivirus at a titer of 10^−9^ TU/mL. The injections of chickens from different groups were all performed using a 0.45 mm diameter needle for wing vein injection. The chickens were allowed free access to food and water. After 7 days, they were slaughtered, and the liver, heart, and pectoral muscle tissue samples were collected for RNA transcriptome sequencing.

The gene expression data for KLF family members were screened using transcriptome data from liver, heart, and muscle tissues treated with miR-22. The mapped read counts and transcript lengths in the samples were then standardized and normalized. The fragments per kilobase of transcript per million mapped fragments (“FPKM”) method was applied for this calculation, with the formula as follows:$$FPKM=\frac{cDNA\;Fragments}{Mapped\;Fragments\left(Millions\right)\;*\;Transcript\;Length\;(kd)}$$

FPKM calculation formula: (cDNA fragments represent the number of fragments that are aligned to a transcript; mapped fragments (millions) indicate the total number of fragments aligned to the transcript, in 10^6^; Transcript length (kb): length of the transcript, in units of 10^3^ bases.)

Gene expression heatmaps were generated using FPKM values from the *KLF* gene family. The resulting heatmaps were then analyzed to assess the outcomes.

### 2.8. Statement of Ethics

This study took place at Foshan University, where the animal protocol was reviewed and approved by the institution’s Experimental Animal Welfare and Animal Experiment Ethics Review Committee (approval number: FOSU202103-28). The procedures adhered to the guidelines set by the Animal Use Committee of the Ministry of Agriculture of China, Beijing, which are designed to minimize animal suffering throughout the research process.

## 3. Result

### 3.1. Identification and Physicochemical Properties of KLF Gene Family Members in Chicken

By conducting sequence homology alignment with pigs, frogs, and zebrafish, 14 *KLF* family members were identified from the chicken genome library, named *KLF1-KLF13* and *KLF15* (Table 1). The protein amino acid length range in the *KLF* gene family is 235 to 530, with a molecular weight ranging from 25,661.19 to 56,922.34 Da. The theoretical isoelectric point (PI) ranges from 6.5 to 9.54. These family members are unevenly distributed across 10 chromosomes, with KLF11 having the longest coding amino acid sequence of 530 amino acids and KLF9 having the shortest, with only 235 amino acids. Additionally, KLF15 has the most exons, with 14, while KLF1 has the fewest, with only one. It is worth noting that KLF1 (LOC424577), located on chromosome 8, is described as “Krüppel-like factor 1 (erythroid)-like” and is also referred to as KLF-4 or KLF1. In contrast, KLF4, whose full name is “Krüppel-like factor 4”, is located on the Z chromosome. These two are distinct genes.

### 3.2. Chromosomal Location of KLF Gene Family

Through chromosome mapping analysis, it was found that the 14 members of the *KLF* gene family are unevenly distributed across 10 chromosomes (Figure 1). Two *KLF* family genes were found on chromosomes 1, 2, 4, and Z, while one KLF family member was found on chromosomes 3, 7, 8, 10, 12, and 28. No members of the *KLF* gene family were found on other chromosomes.

### 3.3. Phylogenetic Analysis of the KLF Gene Family

To better reveal the evolutionary relationship of the *KLF* gene family, 63 protein sequences of *KLF* genes were collected from four species, including *Gallus gallus* (chicken), *Sus scrofa* (pig), *Xenopus tropicalis* (tropical clawed frog), and *Danio rerio* (zebrafish). Using the NJ method in the MEGA7 software (v7.0.26), a systematic evolutionary tree was constructed (Figure 2). Based on the tree, the KLF family can be divided into two subfamilies, where KLF1-8, KLF12, and KLF17 make up subfamily I, while the remaining family members make up subfamily II. Each subfamily can be further divided into several subgroups. For example, subfamily I includes the subgroups KLF5/6/7, KLF3/8/12, and KLF1/2/4/17. Subfamily II includes the subgroups KLF10/11, KLF14/16, KLF9/13, and KLF15.

### 3.4. Gene Structure and Motif Sequence Analysis of KLF Gene Family

The gene structure of the KLF family members was visualized using an exon–intron organization diagram (Figure 3). The length of KLF gene sequences ranges from 1923 to 262,674 bp, with KLF2 having the shortest and KLF12 having the longest sequence. The number of exons in the 14 KLF genes ranges from 1 to 15, with KLF14 having the highest number of exons and KLF1 having the lowest.

### 3.5. Motif Sequence Analysis of KLF Gene Family

The *KLF* gene family sequences were analyzed using the online tool MEME, which identified 10 motifs named motif 1 to motif 10. The analysis, shown in Figure 4, revealed that the 3’ end of the 1–14 members of the *KLF* gene family contains four identical motifs (motif 1, motif 2, motif 3, and motif 5) arranged in a consistent order. The KLF family’s main structural feature is the zinc finger domain located at the carboxy-terminal end, indicating that motifs 1/2/3/5 form the structural domain of this family.

### 3.6. Gene Collinearity Analysis of Chicken KLF Family

The Circos plot in Figure 5 illustrates the number and location of *KLF* gene family members on each chromosome. In chickens, 14 *KLF* genes are located at various positions across 10 different chromosomes, with each *KLF* gene having one or more predicted or validated transcripts (specific transcripts corresponding to each *KLF* gene are detailed in Appendix A). The innermost circle and the middle circle of the plot represent the gene density on the chromosomes using heatmaps and curves, respectively, indicating that these chromosomes have relatively low gene density. The syntenic relationships among the *KLF* genes are as follows: *KLF5* and *KLF12* on chromosome 1 exhibit synteny with *KLF8* on chromosome 4; *KLF8* on chromosome 4 also exhibits synteny with *KLF3* on the same chromosome. Additionally, *KLF6* on chromosome 2 shows synteny with *KLF7* on chromosome 7, while *KLF10* on chromosome 2 is syntenic with *KLF11* on chromosome 3. Furthermore, *KLF13* on chromosome 10 and *KLF9* on the Z chromosome exhibit synteny. No syntenic relationships were observed among the other *KLF* genes.

### 3.7. Prediction of the Tertiary Structure and Transmembrane Region of KLF Family Proteins

During the search for homologous three-dimensional structure templates, it was found that the template 5wjq.1.C had the highest global model quality estimation value (GMQE) for predicting the three-dimensional structure of KLF9 protein, which was 0.28. This indicates that template 5wjq.1.C is the best template. The predicted tertiary structure of the KLF9 protein is shown in Figure 6. For the other 13 members of the chicken *KLF* gene family, the protein tertiary structures predicted using their respective appropriate templates are shown in Appendix A (A~M). Using template 5wjq.1.C, the protein tertiary structures of KLF9 in fish (zebrafish), amphibians (tropical clawed frogs), birds (chickens), and mammals (pigs) were predicted, and the coverage of their protein sequences on the template were 34.21%, 37.37%, 39.80%, and 40%, respectively.

The online tools TMHMM (https://services.healthtech.dtu.dk/service.php? TMHMM-2.0, accessed on 22 November 2022) were used to analyze the transmembrane region protein sequences of chicken *KLF* gene family members. The probability of all family members crossing the transmembrane region is not close to 1, which is indicated by the output purple line. For example, the transmembrane regions of protein sequences for KLF9 (Figure 7) were predicted using the online tool TMHMM. Appendix A (A~M) shows the predicted transmembrane structure of protein sequences for another 13 members of the chicken *KLF* gene family. The results show that the proteins of all members have no transmembrane sequence. Therefore, it is speculated that all members of the chicken *KLF* gene family belong to secretory proteins.

### 3.8. Effect of miRNA-22 on the Expression of KLF Gene Family in Different Tissues of Qingyuan Partridge Chickens

To investigate the effects of lentivirus-mediated miR-22 knockdown or overexpression on the expression of KLFs in the liver, heart, and muscle tissues of Qingyuan partridge chickens, transcriptome sequencing was conducted. The results are presented in Figure 8a–c. Figure 8a illustrates the expression levels of KLFs in the liver following miR-22 overexpression and interference. Specifically, *KLF5* and *KLF7* exhibited low expression across all groups, while *KLF10* and *KLF15* were highly expressed. Notably, miR-22 significantly affected the expression of *KLF3*, *KLF6*, and *KLF12* in the liver compared to the control group (*p* < 0.05). However, there were no significant changes in the expression of *KLF2*, *KLF5*, *KLF7*, *KLF9*, *KLF10*, *KLF11*, *KLF13*, and *KLF15* (*p* > 0.05). Figure 8b shows the impact of miR-22 interference on the KLF expression in the heart. The results indicate that *KLF3*, *KLF5*, *KLF6*, *KLF7*, *KLF9*, *KLF11*, and *KLF12* were expressed at very low levels in the heart, whereas *KLF2*, *KLF10*, *KLF13*, and *KLF15* exhibited higher expression levels, albeit with considerable individual variability within groups. Figure 8c depicts the effect of miR-22 interference on KLFs expression in the pectoral muscle. Similar to the heart, *KLF2*, *KLF3*, *KLF5*, *KLF7*, *KLF11*, and *KLF12* showed very low expression levels in the pectoral muscle, while *KLF9*, *KLF10*, *KLF13*, and *KLF15* were more highly expressed, though with significant individual variability within groups. It is worth noting that KLF1, KLF4, and KLF8 were undetectable in all three tissues.

## 4. Discussion

In the past few decades, there has been a significant amount of research on the biological functions of the mammalian *KLF* gene family, such as promoting the formation of fat and muscle, participating in tumorigenesis, and regulating the metabolism of cells and tissues. However, there has been relatively little research on the poultry side, and the functions and mechanisms of many family members have yet to be explored. The Krüppel-like factors family (KLF) is widely present in eukaryotes and is a family of zinc finger protein transcription factors that can bind to DNA. They play an important role in lipid metabolism and can regulate the differentiation of adipocytes. In this study, 14 *KLF* genes were identified in the chicken genome. Through analysis of their physicochemical properties, conserved motifs, and sequence homology, it was found that KLF9 and KLF13, belonging to the same subgroup, have 235 and 277 amino acids, respectively, a theoretical isoelectric point of 9.44 and 9.54, and contain the same motif with the highest sequence homology. It is speculated that these two genes may have a similar function, which is also the case in subgroups KLF3/8/12.

Chromosome localization analysis was conducted on members of the *KLF* gene family, and it was found that the 14 members were unevenly distributed on 10 chromosomes. Among them, *KLF3* and *KLF8* are both located on chromosome 4. By comparing the physicochemical properties and gene structures of the two, it was found that they have similar amino acid lengths and intron–exon structures, which indirectly confirms that they may have a close evolutionary relationship.

Based on the branching of the constructed phylogenetic tree of the system (Figure 2), the chicken *KLF* gene family can be divided into two subfamilies, I and II. Previous studies have shown that members of the chicken *KLF* family, including *KLF2* [28], *KLF3* [14], *KLF5* [29], and *KLF7* [16], play a positive regulatory role in adipogenesis, while family members *KLF11* [17] and *KLF15* [15] inhibit adipogenesis. Interestingly, the family members that play a positive regulatory role in adipogenesis belong to subfamily I, while those that play a negative regulatory role belong to subfamily II. This suggests that the members of subfamily I may all play a positive regulatory role in adipogenesis, while those of subfamily II play a negative regulatory role. This finding provides some insights for subsequent relevant research. Within the same subfamily, the members of the family have high sequence homology, indicating that they are likely replicated from a common ancestral gene. Interestingly, chicken KLF1 is very close to tropical clawed frog and zebrafish KLF17 on the evolutionary tree, which suggests that the true identity of the gene currently named chicken *KLF1* is most likely *KLF17*, consistent with the research findings of Antin et al. [4].

The evolution of species has undergone a process of transition from fish to amphibians, then to birds, and finally to mammals. In the constructed phylogenetic tree, the KLF factors of avian species are closer to those of amphibians and mammals, which is consistent with the evolutionary history of species. Upon closer observation of the evolutionary tree, it is found that in subfamily II, the KLF family members of pig, a mammal, have two more members than other species (zebrafish, tropical clawed frog, chicken), *KLF14* and KLF16. Some researchers [30] believe that this is a result of genome duplication, where *KLF16* is replicated from *KLF9* and *KLF13*. Motif analysis of the KLF family reveals that only KLF5 lacks motif4 in subgroups 5/6/7, indicating that the motif of KLF5 may have been lost during the evolutionary process.

The presence of synteny between *KLF* genes suggests they may share similar functions in coordinating signaling pathways. As depicted in Figure 5 of this study, *KLF3* and *KLF8*, as well as *KLF5* and *KLF12*, exhibit significant synteny, indicating potential roles in synergistically regulating the same pathways. Further examination of the phylogenetic tree in Figure 2 reveals that these genes belong to the same subfamily, demonstrating high homology. Previous research has shown that *KLF3* is involved not only in regulating fat formation and lipid metabolism in chickens [31] but also influences the proliferation and differentiation of skeletal muscle satellite cells (SMSCs) [32]. Similarly, studies have linked *KLF5* to the proliferation, differentiation, and apoptosis of chicken preadipocytes [19], as well as to fat generation and metabolism [31], and SMSCs activities [32,33]. Additionally, *KLF5* is speculated as participating in regulating chicken skeletal muscle development [34]. Research on *KLF8* and *KLF12* in poultry is less extensive. Studies have associated *KLF8* and *KLF12* with the differentiation of goat preadipocytes [35,36]. Ling et al. also noted the differential expression of *KLF12* before and after differentiation of chicken preadipocytes [37]. Moreover, *KLF8* is implicated in regulating cardiac disease, cancer, neuronal manipulation, and brain homeostasis [38,39,40]. Simultaneously, there exists synteny between *KLF6* and *KLF7*, *KLF10* and *KLF11*, as well as *KLF9* and *KLF13* genes. Interestingly, these paired genes showing synteny are closely clustered in the phylogenetic tree and belong to the same subfamily. Studies have revealed specific functions for these genes across various physiological contexts. *KLF6* gene expression decreases with age in the pectoral muscles of Tibetan chickens, while in leg muscles, its expression initially increases with age before declining [41]. *KLF7* has been shown to inhibit the differentiation of chicken preadipocytes and promote their proliferation [42]. Notably, associations with obesity-related SNPs have been reported separately for *KLF6* and *KLF7* [43,44]. *KLF10* effectively inhibits the proliferation of chicken myoblasts [45], while *KLF11* is implicated in duck muscle development [46] and is associated with cartilage cell development and osteogenesis in disease contexts. Both *KLF10* and *KLF11* are closely linked to hepatic gluconeogenesis, hyperglycemia, glucose intolerance, diabetes, and hepatic metabolic disorders [47,48]. The research on *KLF13* in poultry is limited, but in pigs, *KLF13* has been shown to significantly affect the development of fat cells. miR-125a-5p enhances the proliferation and inhibits the differentiation of pig intramuscular adipocytes by regulating KLF13 [49]. KLF13 promotes the differentiation of pig fat cells by activating PPARγ [50]. Moreover, KLF9 regulates the differentiation of fat cells by acting on the PPARγ promoter [51]. Both *KLF9* and *KLF13* inhibit the activation of the AKT signaling pathway, thereby suppressing the growth of prostate cancer cells [52,53]. In summary, the synteny observed between *KLF3* and *KLF8*, *KLF5* and *KLF12* likely contributes to their collective influence on adipocyte differentiation and skeletal muscle satellite cell proliferation. The synteny between *KLF6* and *KLF7* may collectively regulate intramuscular fat deposition and obesity in the organism. The coordinated regulation of *KLF10* and *KLF11* manifests in bone cell development, bone diseases, and certain metabolic disorders. *KLF9* and *KLF13* both regulate the differentiation and proliferation of fat cells internally and can serve as potential therapeutic targets for prostate cancer cell proliferation.

Based on the protein sequences of KLF family members, the protein tertiary structures of each member were predicted using the online software SWISS-MODEL (https://swissmodel.expasy.org/, accessed on 28 October 2022). The GMQE value of KLF9 protein tertiary structure was found to be the highest among all members when using 5wjq.1.C as the template, indicating that the protein structure prediction result of KLF9 was more reliable compared to other members. By comparing the predicted protein tertiary structures of KLF9 in zebrafish, tropical clawed frog, chicken, and pig based on the template 5wjq.1.C, it was found that the similarity of KLF9 protein tertiary structure was higher between chicken and mammalian pig, suggesting that KLF9 may perform similar functions in chicken and pig. In addition, transmembrane region prediction was performed on the protein sequences of all members of the KLF family in chicken, and it was found that all members of this family belong to non-transmembrane proteins. The results above provide a scientific foundation for studying the chicken *KLF* gene family.

In summary, the functions of the *KLF* family genes include the regulation of intramuscular fat deposition, adipocyte differentiation, and skeletal muscle cell proliferation. Previous studies have demonstrated that miR-22-3p plays a regulatory role in fat deposition, skeletal muscle cell proliferation, and cardiomyocyte apoptosis. For instance, miR-22-3p is most abundantly expressed in the liver of laying poultry, where it affects hepatic fat synthesis. Additionally, miR-22-3p is a significant factor influencing the proliferation and differentiation of primary skeletal muscle cells in Hu sheep. It is also notably upregulated in cardiomyocytes and plasma exosomes of mice with chronic myocardial infarction and patients with heart failure. Both the *KLF* family genes and miR-22-3p are involved in physiological processes such as intramuscular fat deposition, adipocyte differentiation, skeletal muscle cell proliferation, and cardiac diseases. This raises the question: what impact does miR-22-3p have on the expression of *KLF* family genes? Using transcriptome sequencing technology, we analyzed the expression levels of *KLF* family genes in the liver, heart, and pectoral muscles of Qingyuan partridge chickens after miR-22 interference or overexpression. Studies have found that miR-22 influences the expression of *KLF3*, *KLF6*, and *KLF2* in the liver, suggesting that miR-22 may directly or indirectly regulate the expression of these KLFs. Additionally, *KLF10* and *KLF15* were highly expressed in the liver, heart, and pectoral muscles, with significant differences observed in some tissue samples. Parakati et al. [45] demonstrated that *KLF10* can inhibit myoblast proliferation by suppressing the activity of the fibroblast growth factor receptor 1 (FGFR1) promoter. Raza et al. [54] found that *KLF10* is associated with liver metabolism and fat generation. *KLF15* is an important regulator of lipid metabolism in adipocytes. Studies have confirmed that the *KLF15* gene is related to abdominal fat deposition in poultry [18], growth traits and carcass characteristics [55], and skeletal muscle atrophy in chickens [34]. These findings suggest that miR-22-3p may influence intramuscular fat deposition, myoblast proliferation, and carcass growth by directly or indirectly regulating *KLF10* and *KLF15*. These results provide a valuable reference for future research, although the underlying molecular regulatory mechanisms require further investigation.

## 5. Conclusions

In this study, 14 *KLF* genes in chicken were identified. Chromosomal localization results indicate that these 14 *KLF* genes are unevenly distributed across 10 different chromosomes. Phylogenetic analysis conducted on representative species from fish, amphibians, birds, and mammals suggests that the *KLF* genes can be divided into two categories. Based on evolutionary relationships and the phylogenetic tree, the KLF family in birds is more closely related to those in mammals and amphibians than to those in fish. KLFs within the same subgroup are likely derived from a common ancestral gene, indicating that members such as *KLF3*, *KLF8*, and *KLF12* within the same subgroup may have similar or overlapping functions. Predictions of protein tertiary structure and transmembrane regions reveal that all chicken KLF family members are non-transmembrane proteins. Interference and overexpression of miRNA-22 were found to affect the expression levels of certain *KLF* genes in the liver, heart, and muscles of Qingyuan partridge chickens. This study provides scientific evidence of the structure of the *KLF* gene family and their expression in chicken liver, heart, and muscles, offering theoretical insights for further research on the *KLF* gene family.

## Figures and Tables

**Figure 1 animals-14-02594-f001:**
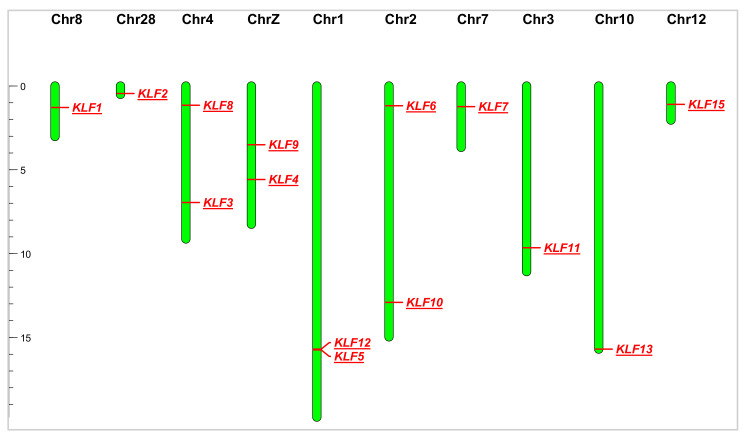
The distribution of chicken *KLF* gene family members on chromosomes.

**Figure 2 animals-14-02594-f002:**
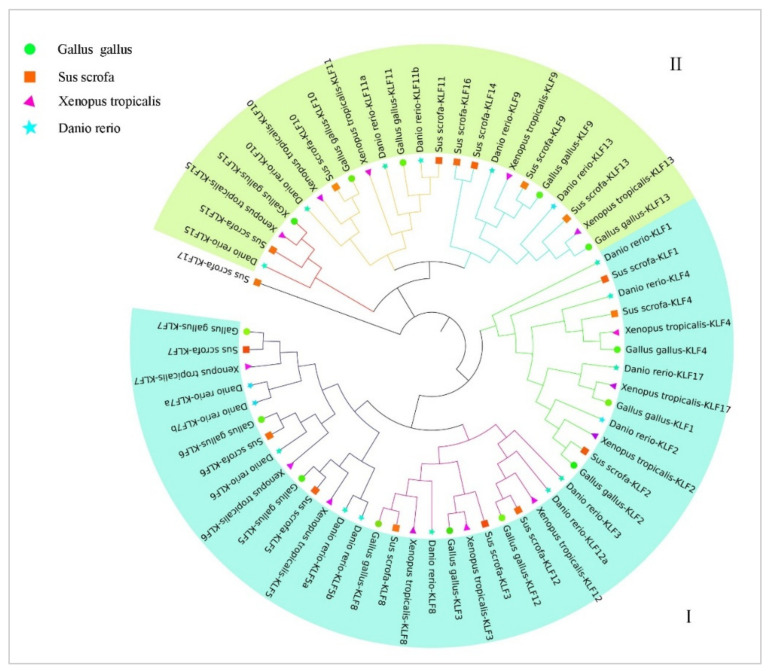
The unrooted phylogenetic tree of the *KLF* gene family in *Gallus gallus* (chicken), *Sus scrofa* (pig), *Xenopus tropicalis* (tropical clawed frog), and *Danio rerio* (zebrafish).

**Figure 3 animals-14-02594-f003:**
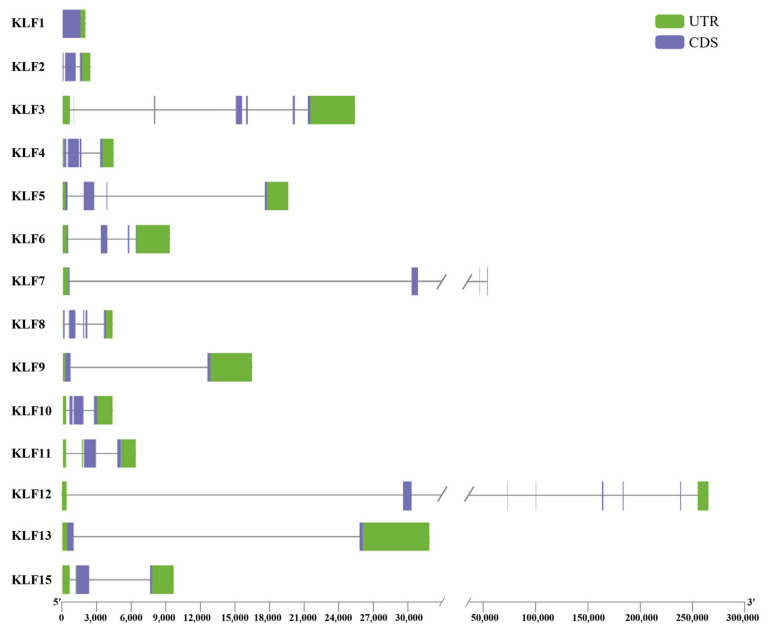
The gene structure of *KLF* genes in chicken. The green boxes and perple boxes in the gene structure diagram represent UTR and CDS.

**Figure 4 animals-14-02594-f004:**
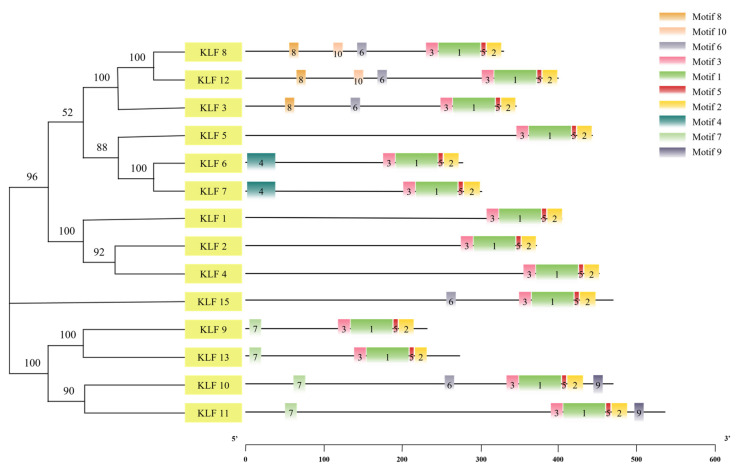
Phylogenetic relationships and conserved motif of *KLF* genes in chicken.

**Figure 5 animals-14-02594-f005:**
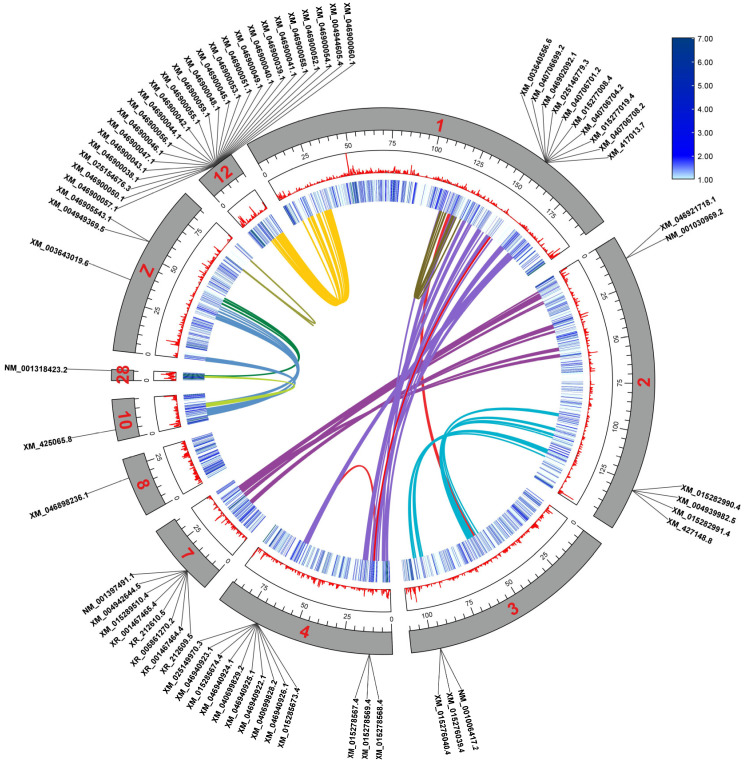
Collinearity between chicken *KLF* gene families. The outermost ring displays the names of predicted or validated transcripts for each gene. The gray bands in the outer ring represent chromosomes, with the red numbers or letters inside indicating chromosome numbers. The middle ring, featuring red peaks and blue bands, represents gene density on each chromosome. The innermost layer shows colored lines representing the synteny relationships between KLF genes. The top right section indicates the gene density bands.

**Figure 6 animals-14-02594-f006:**
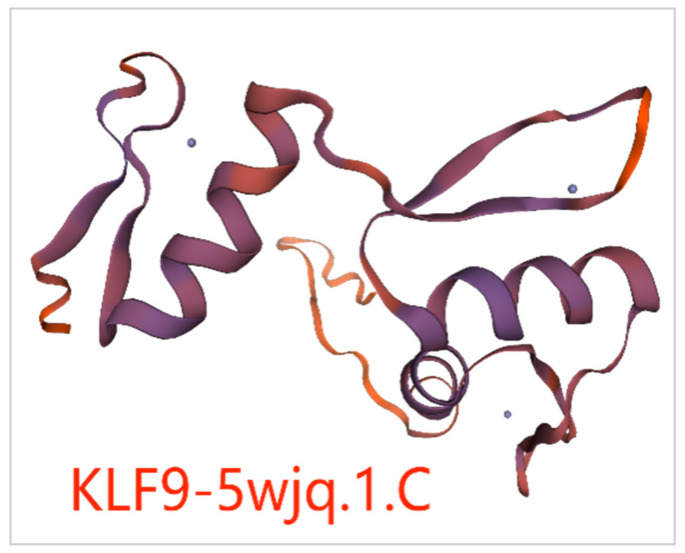
Prediction of KLF9 protein tertiary structure.

**Figure 7 animals-14-02594-f007:**
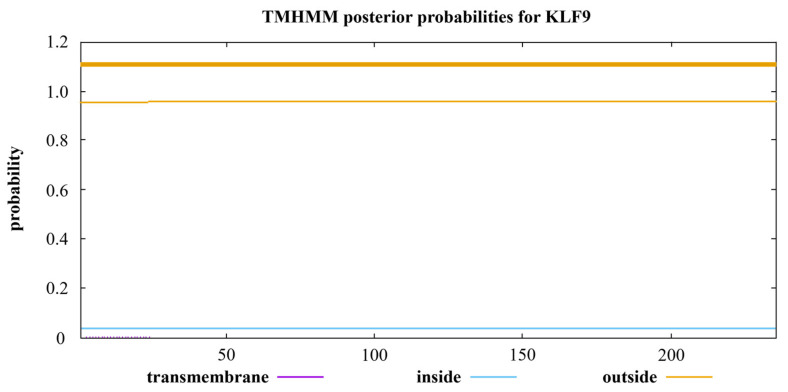
Prediction of KLF9 protein transmembrane structure.

**Figure 8 animals-14-02594-f008:**
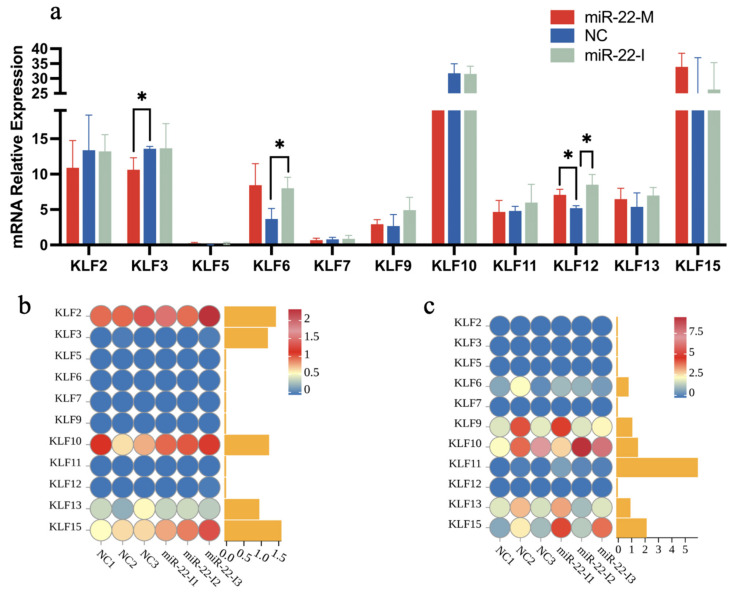
The impact of miR-22 on *KLF* gene relative expression in different tissues. (**a**) Shows the expression of KLFs in the liver after lentiviral miR-22 interference (miR-22-I) or overexpression (miR-22-M); (**b**) depicts the expression of KLFs in the heart following lentiviral miR-22 interference (miR-22-I); (**c**) illustrates the expression of KLFs in the pectoral muscle after lentiviral miR-22 interference (miR-22-I). FPKM represents the normalized expression levels across different samples. Blue indicates lower expression and red indicates higher expression (**b**,**c**). * indicate significant differences between the different groups (n = 3, *p* < 0.05, Student’s t-test).

**Table 1 animals-14-02594-t001:** Physicochemical properties of chicken *KLF* gene family proteins.

Gene Name	Gene ID	Chr	Genomic Location	Exon	AA	MW (Da)	PI
*KLF1*	424577	8	20,104,457–20,106,379	1	501	52,966.89	7.33
*KLF2*	420148	28	4,830,902–4,833,231	3	380	42,069.22	7.01
*KLF3*	429811	4	69,048,099–69,075,232	11	347	39,027.04	9.53
*KLF4*	770254	Z	56,393,942–56,398,448	5	458	48,565.25	8.63
*KLF5*	418818	1	156,007,859–156,026,745	4	441	48,836.51	8.63
*KLF6*	420463	2	11,740,170–11,749,143	4	283	31,965.53	6.50
*KLF7*	429011	7	11,883,355–11,944,567	5	296	32,981.18	7.98
*KLF8*	101749773	4	11,420,793–11,426,412	7	330	35,577.55	8.75
*KLF9*	770238	Z	35,630,341–35,646,189	2	235	25,661.19	9.44
*KLF10*	420255	2	128,801,525–128,806,420	5	466	50,178.02	9.41
*KLF11*	421934	3	96,177,645–96,185,317	5	530	56,922.34	9.11
*KLF12*	418817	1	155,542,183–155,804,856	12	396	43,470.94	9.49
*KLF13*	427493	10	5,587,589–5,618,322	2	277	30,776.50	9.54
*KLF15*	427588	12	10,806,952–10,841,285	14	463	51,113.08	8.71

Note: Exon, AA, MW, and PI denote the exon number, amino acid count, molecular weight, and isoelectric point, respectively.

## Data Availability

The original contributions presented in the study are included in the article/Appendix A; further inquiries can be directed to the corresponding author/s.

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
