# Peer review of "Genome-Wide Identification, Evolution, and miRNA-22 Regulation of Kruppel-Like Factor (KLF) Gene Family in Chicken (Gallus gallus)"

_animals, 2024, doi:10.3390/ani14172594_

Round 1
Reviewer 1 Report
Comments and Suggestions for Authors
The objective of the manuscript, entitled "Genome-wide Identification, Evolution, and miRNA-22 Regulation of Kruppel-Like Factor (KLF) Gene Family in Chicken (Gallus gallus)", is to provide a comprehensive identification and analysis of the KLF gene family in chickens through the utilisation of bioinformatics tools. The study investigates the evolutionary relationships of KLF genes across various species, characterises their structural features, and examines the regulatory impact of miRNA-22 on KLF gene expression in different chicken tissues. The paper's principal contributions are the identification of 15 KLF family members in chickens, insights into their evolutionary proximity to mammalian KLFs, and the discovery of tissue-specific effects of miRNA-22 on KLF expression. These findings establish a foundation for future research on the functional roles of KLF genes in chickens, particularly in relation to lipid metabolism and tissue differentiation.
The manuscript offers valuable insights into the KLF gene family in chickens; however, several areas require improvement to enhance its overall quality and impact.
The bioinformatics analysis is comprehensive in its identification and characterisation of KLF genes; however, it lacks depth in terms of functional validation. The study is primarily based on in silico predictions, with a lack of sufficient experimental verification of the predicted functions of the KLF genes. This limitation also extends to the investigation of the regulation of KLF genes by miRNA-22, which, while intriguing, lacks mechanistic insights. To enhance the study, it would be beneficial to conduct more detailed functional assays, such as luciferase reporter assays, to confirm the direct miRNA-22 targets among the KLF genes.
A notable shortcoming is the absence of comprehensive statistical analyses and validation of the bioinformatics predictions. The study fails to adequately address the statistical significance of the observed expression changes in KLF genes upon miRNA-22 interference and overexpression. The reliability of the findings could be enhanced by the inclusion of robust statistical tests and the addressing of multiple testing corrections.
The experimental design, in particular the sample size, is not sufficiently robust. The study employs a relatively modest sample size of nine chickens, divided into three groups, which may not afford sufficient statistical power to draw definitive conclusions. It would be beneficial to increase the sample size in order to improve the robustness of the experimental results.
Here are specific comments referring to line numbers, tables, or figures that point out inaccuracies or unclear content within the scientific text:
Lines 117-125: The methods for identifying KLF family members are not sufficiently detailed. Specific parameters for the HMM search and BLAST comparisons should be provided to ensure reproducibility.
Lines 137-143: The phylogenetic analysis lacks justification for the choice of species used (zebrafish, frogs, chicken, and pigs). An explanation for why these particular species were selected would strengthen the evolutionary context.
Lines 163-170: The experimental design for the miRNA-22 manipulation study is unclear. More details on the lentiviral constructs, injection method, and sample size justification are needed.
Lines 173-182: The explanation of the CPM normalization method is confusing. A clearer description of how this normalization accounts for both transcript length and sequencing depth would be beneficial.
Lines 186-188: The sudden introduction of miR-122-5p target gene prediction seems out of place, as the study focuses on miR-22-3p. This inconsistency needs to be addressed or explained.
Addressing these areas of improvement would considerably enhance the robustness and impact of the study, thereby making a more substantial contribution to the field of animal genomics and epigenetics. Overall, the manuscript can be recommended for publication once the problems in individual lines have been corrected. I suggest that conceptual changes should be made when writing further reports on the KLF gene family study in chickens.
Reviewer 2 Report
Comments and Suggestions for Authors
The research undertaken by the authors of the manuscript is interesting with significantly aspects of novelty and follows the current research trend in this field. I believe that the experiment was properly carried out using appropriate material and research methods. The obtained results are interesting and valuable. I just have a few minor comments and suggestions:
Simple Summary
Line 12 – The sentence states that fish, amphibians, birds, and mammals are species, and we know that they are classes; please clarify.
INTRODUCTION
I suggest expanding on the abbreviations first used throughout the chapter.
Lines 43-44 – I believe that the phrase “zinc finger proteins” should be removed from the sentence or the entire sentence should be reworded.
Lines 47-48 – There are sources that mention 18 members of the KLF protein family. Please verify.
Line 97 – “Vanessa M.” should be removed from the sentence.
Lines 107-115 – I think this fragment should be reworded to make the purpose of the work clear.
MATERIALS AND METHODS
Line 153 – I believe that “GXF gene Position & inf function” should be changed to “GXF Gene Position & Info. extract function".
Line 163 – I think the subtitle needs to be reworded.
Lines 184-185 – I think this part should be reworded so that the sentence doesn't sound like a command.
RESULTS
Line 208 and table 1 – Please verify whether the given values ​​are in kDa or Da.
Line 215 – I think the entry should be deleted.
Line 226 - The use of "frog" seems too general (taxonomically this name designates a family, not a species); I believe that it would be justified to provide the name of the studied species, i.e. tropical clawed frog; this remark also applies to the rest of the text (lines 235, 280, 368 and 420).
Line 237 – Please change “TThe” to “The”.
Lines 239-243 and Figure 3 - In the figure, KLF2 appears to have the shortest sequence, and the description states that KLF1 has the shortest sequence. It is also difficult to comment on the number of exons. Moreover, "KLF14" should be changed to "KLF15".
Lines 266-267 – I believe that the information should be supplemented with the chromosomal location of the KLF13 gene.
Line 284 – Please verify the entry "transmembrane"; in my opinion it should be "tertiary".
Lines 297-298 – In my opinion, the subtitle needs to be reworded.
Line 327 – I think the "is illustrated in the figures" entry is unnecessary.
DISCUSSION
There are studies in which the chicken KLF4 gene was mapped to chromosome 8. Additionally, NCBI resources include information on LOC424577 (Kruppel-like factor 1 (erythroid)-like), also known as: KLF-4, KLF1. If the results of the presented research provide some order, it may be worth mentioning it.
Line 342 – I believe “and KLF9/13” is a repeat of an earlier description, so it should be removed.
Line 362 – “Parker B.” should be deleted and "et al." should be added after the name Antin.
Line 443 – “Rajini” should be deleted.
Line 444 – “Sayed Haidar Abbas” should be deleted.
Line 444 – The abbreviation “FGFR1” can be expanded.
Lines 453-466 – This paragraph summarizes the research. Please consider moving individual fragments of this paragraph to other places in the manuscript, e.g. part to the beginning of the DISCUSSION chapter and part to the CONCLUSIONS chapter.
Round 2
Reviewer 1 Report
Comments and Suggestions for Authors
I am grateful for the opportunity to review the authors' responses and the revised manuscript. I am glad to the authors for their efforts in addressing the comments and suggestions provided in the initial review.
1. The authors have acknowledged the necessity for functional validation of the KLF gene family and the relationship between miRNA-22 and KLF genes. It is commendable that the authors have expressed a commitment to conducting further functional assays in future studies, which demonstrates an understanding of the importance of experimental validation in supporting their bioinformatics findings.
2. In regard to the statistical analyses, the authors have implemented notable enhancements by normalising the KLF gene expression data to FPKM values in lieu of CPM. This modification improves the precision of the results by incorporating data on sequencing depth and gene length. Furthermore, the authors have revised the figures to emphasise significant differences, thereby enhancing the presentation of their findings.
3. Furthermore, the authors have addressed the concerns regarding sample size, elucidating the constraints imposed by the high cost of experiments involving lentivirus. It is acknowledged that budgetary constraints may impact sample sizes; however, the implications of this limitation on the findings and the necessity for larger sample sizes in future studies to enhance statistical power should be emphasised.
4. In response to requests for more detailed methods to identify KLF families, the authors have added specific parameters for HMM searches and BLAST comparisons, improving the reproducibility of their work. Similarly, the rationale for the choice of species in the phylogenetic analysis has been clarified, providing a better understanding of the evolutionary context.
5. The authors have also improved the clarity of their experimental design for the miRNA-22 manipulation study by providing additional details on lentiviral constructs and injection methods. This improvement addresses previous concerns and strengthens the overall methodological rigour of the study.
6. The authors have acknowledged and corrected the confusion surrounding the normalisation method by switching to FPKM to allow more accurate comparisons of gene expression levels. This change is a positive step towards improving the clarity of the methodology.
7. Furthermore, the removal of the redundant paragraph regarding miR-122-5p target gene prediction is appreciated, as this aligns the focus of the study more closely with miR-22-3p.
Thus, the authors have made commendable efforts to address the weaknesses identified in the initial review, resulting in a more robust manuscript. The revisions improve the clarity, reproducibility and overall quality of the study. I appreciate the authors' responsiveness to the feedback provided and their commitment to improving their work. I believe that the manuscript is now suitable for publication. Thank you for the opportunity to review this revised manuscript.